# LEARNING TO LEARN WITH GENERATIVE MODELS OF NEURAL NETWORK CHECKPOINTS

## ABSTRACT

We explore a data-driven approach for learning to optimize neural networks. We construct a dataset of neural network checkpoints and train a generative model on the parameters. In particular, our model is a conditional diffusion transformer that, given an initial input parameter vector and a prompted loss, error, or return, predicts the distribution over parameter updates that achieve the desired metric. At test time, it can optimize neural networks with unseen parameters for downstream tasks in just one update. We find that our approach successfully generates parameters for a wide range of loss prompts. Moreover, it can sample multimodal parameter solutions and has favorable scaling properties. We apply our method to different neural network architectures and tasks in supervised and reinforcement learning.

## 1 INTRODUCTION

Gradient-based optimization is the fuel of modern deep learning. Techniques of this class, such as SGD (Robbins & Monro, 1951) and Adam (Kingma & Ba, 2015), are easy to implement, scale reasonably well and converge to surprisingly good solutions—even in high-dimensional, non-convex neural network loss landscapes. Over the past decade, they have enabled impressive results in computer vision (Krizhevsky et al., 2012; Girshick et al., 2014), natural language processing (Vaswani et al., 2017; Radford et al., 2018) and audio generation (Van Den Oord et al., 2016).

While these *manual* optimization techniques have led to large advances, they suffer from an important limitation: they are unable to improve from past experience. For example, SGD will not converge any faster when used to optimize the same neural network architecture from the same initialization the 100th time versus the first time. *Learned* optimizers capable of leveraging their past experiences have the potential to overcome this limitation and may accelerate future progress in deep learning.

Of course, the concept of learning improved optimizers is not new and dates back to the 1980s, if not earlier, following early work from Schmidhuber (1987) and Bengio et al. (1991). In recent years, significant effort has been spent on designing algorithms that learn via nested meta-optimization, where the inner loop optimizes the task-level objective and the outer loop learns the optimizer (Andrychowicz et al., 2016; Li & Malik, 2016; Finn et al., 2017). In some instances, these approaches outperform manual optimizers. However, they are challenging to train in practice due to a reliance on unrolled optimization and reinforcement learning.

Taking a modern deep learning perspective suggests a simple, scalable and data-driven approach to this problem. Over the past decade, our community has trained a massive number of checkpoints. These checkpoints contain a wealth of information: diverse parameter configurations and rich metrics such as test losses, classification errors and RL returns that describe the quality of the checkpoint. Instead of leveraging large-scale datasets of images or text, we propose learning from large-scale datasets of *checkpoints* recorded over the course of many training runs.

To this end, we create a dataset of neural network checkpoints (Figure 1, left). Our dataset consists of 23 million checkpoints from over a hundred thousand training runs. We collect data from supervised learning tasks (MNIST, CIFAR-10) as well as reinforcement learning tasks (Cartpole), and across different neural network architectures (MLPs, CNNs). In addition to parameters, we record relevant task-level metrics in each checkpoint, such as test losses and classification errors.

---

Please see our project website in supplementary materials for additional results and visualizations.

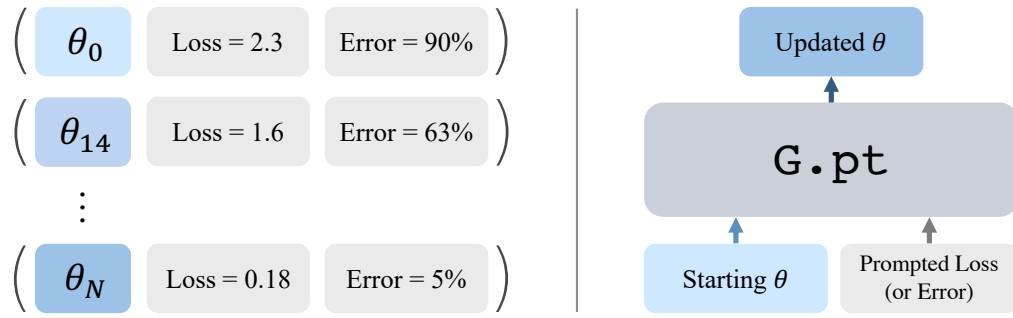

Figure 1: **Generative pre-training from checkpoints.** *Left:* We build a dataset of neural network checkpoints from many training runs. Each checkpoint includes the neural network's parameters and relevant metadata (test losses and test errors for supervised learning tasks, returns for RL tasks). *Right:* G.pt, a generative model of checkpoints. G.pt takes a parameter vector and a loss/error/return prompt as input and predicts the distribution over updated parameters that achieve the prompt.

Given this data, we explore generative pre-training directly in parameter space (Figure 1, right). Specifically, we train transformer-based diffusion models of neural network parameters. Given an initial input parameter vector and a target loss, error or return, these models are trained to predict the distribution over updated parameter vectors for a single network architecture that achieve the target metric. Our method is trained using standard generative modeling techniques instead of unrolled optimization and reinforcement learning algorithms. We call our model G.pt[1].

We show that our approach has a number of favorable properties. First, it is able to rapidly train neural networks from unseen initializations with just one parameter update (Figure 3). Second, it can generate parameters that achieve a wide range of prompted losses, errors and returns (Figure 5). Third, it is able to generalize to out-of-distribution weight initialization algorithms (Figure 6). Fourth, as a generative model, it is able to sample diverse solutions (Figure 8). Finally, it can optimize non-differentiable objectives, such as RL returns or classification errors.

## 2 GENERATIVE PRE-TRAINING FROM NEURAL NETWORK CHECKPOINTS

We pre-train a generative model G.pt on neural network checkpoints. At test time, we use it to generate parameters for neural networks that solve a downstream task.

### 2.1 A DATASET OF NEURAL NETWORK CHECKPOINTS

In order to train G.pt, we build a dataset of neural network checkpoints. Each checkpoint contains neural network parameters and relevant task-level metrics like train losses, test errors or returns. We use standard optimizers like Adam and SGD with momentum to generate the parameters, and we randomly save a subset of checkpoints from each training run. Our methodology for generating each individual training run is explained in detail in Algorithm 1. See Section 3 for additional details.

**Augmenting datasets of neural networks.** To offset the computational cost of collecting checkpoints, we use data augmentation in neural network parameter space. Given a checkpoint $(\theta, \ell)$, we construct augmented tuples $(\mathcal{T}(\theta), \ell)$, where $\mathcal{T}(\cdot)$ is the parameter-level augmentation. In order for these augmented tuples to be valid, we need $f_{\mathcal{T}(\theta)}(x) = f_\theta(x)$ for all parameter vectors $\theta$ and all inputs to the neural network $x$. One type of augmentation that meets this criteria is *permutation augmentation*. Consider an MLP. If we apply some permutation to the outgoing weights (and biases) of the input layer and to the incoming weights of the next layer, the output of the neural network will be preserved (Roeder et al., 2021; Schürholt et al., 2021). Different permutations can be sampled for each layer up to the output layer. This technique is generic and can be applied to MLPs and CNNs alike. We apply the same permutation to both the input and target parameters during pre-training.

---

[1]G and .pt refer to generative models and checkpoint extensions, respectively.

| **Algorithm 1** Checkpoint Data Generation | **Algorithm 2** Pre-training from Checkpoints |
|---|---|
| 1: **Input:** Dataset or simulator $D$, neural network $f$, loss function $L$, task metric, meta data store $S$. | 1: **Input:** Number of training runs $K$, checkpoint dataset runs $\{S_k\}_{k=1}^{K}$, G.pt, diffusion process length $J$, diffusion cumulative variance schedule $\bar{\alpha}$. |
| 2: **Initialize:** Learnable parameters $\theta$ for $f$ | 2: **Initialize:** Learnable parameters $\phi$ for $G$ |
| 3: **for** $t = 1, 2, ..., N_{\text{iter}}$ **do** | 3: **for** $i = 1, 2, ..., N_{\text{iter}}$ **do** |
| 4:     # Sample a mini-batch of data | 4:     # Sample a mini-batch of data |
| 5:     $\{\text{inputs}, \text{labels}\}_t \sim D$ | 5:     $\{\theta_{t_1}, \theta_{t_2}, \ell_{t_1}, \ell_{t_2}\}_i \sim S_k$ |
| 6:     # Compute the predictions | 6:     # Noise future parameters |
| 7:     predictions $\leftarrow f_\theta(\text{inputs})$ | 7:     $j \sim U(\{1, ..., J\})$ |
| 8:     # Compute the loss | 8:     $\tilde{\theta}_{t_2} \sim \mathcal{N}(\sqrt{\bar{\alpha}_j}\theta_{t_2}, (1 - \bar{\alpha}_j)I)$ |
| 9:     loss $\leftarrow L(\text{predictions}, \text{labels})$ | 9:     # Compute the predictions |
| 10:     # Update the model's parameters | 10:     $\hat{\theta}_{t_2} \leftarrow G_\phi(\tilde{\theta}_{t_2}, \theta_{t_1}, \ell_{t_2}, \ell_{t_1}, j)$ |
| 11:     $\theta_{t+1} \leftarrow \text{update}(\text{loss}; \theta)$ | 11:     # Compute the loss |
| 12:     # Compute the task metric | 12:     loss $\leftarrow ||\hat{\theta}_{t_2} - \theta_{t_2}||_2^2$ |
| 13:     $\ell_t \leftarrow \text{metric}(\text{predictions}, \text{labels})$ | 13:     # Update G.pt's parameters |
| 14:     # Save the checkpoint | 14:     $\phi_{i+1} \leftarrow \text{update}(\text{loss}; \phi)$ |
| 15:     $S \leftarrow S \cup \{\theta_t, \ell_t\}$ | 15: **end for** |
| 16: **end for** | |

## 2.2 GENERATIVE MODELS OF NEURAL NETWORK CHECKPOINTS

Using our dataset of checkpoints, we train a generative model $G$ that learns to rapidly train other neural networks. Specifically, $G$ predicts the distribution of updated parameters $p_G(\theta^*|\theta, \ell^*, \ell)$, where $\theta$ is the starting (potentially random) neural network parameters, $\ell$ is the starting loss/error/return and $\ell^*$ is a user's prompted loss/error/return. Conditioning on $\ell^*$ allows G.pt to learn from checkpoints with good and bad performance alike. In this section, we describe an instantiation of our approach based on diffusion and transformers.

### 2.2.1 PRE-TRAINING OBJECTIVE: DIFFUSION OF NEURAL NETWORK CHECKPOINTS

We use diffusion (Sohl-Dickstein et al., 2015) as our generative pre-training task. Diffusion is a good generative modeling framework for neural network parameters since the number of forward passes required to sample a novel parameter vector is set by the length of the diffusion process $J$ as opposed to the dimensionality of the data. This instantiation of G.pt samples parameters by gradually denoising the future (updated) parameters $\theta^*$.

**Parameterization.** Given an input corrupted with noise, diffusion models can be parameterized to predict either the signal or the noise (Ho et al., 2020; Nichol & Dhariwal, 2021). Prior work in the image domain has shown that noise prediction outperforms signal prediction. We find that signal prediction works better in our setting empirically, and so we parameterize $G$ to output parameters. We use fixed variances as in Ho et al. (2020).

**Training.** Our model takes two parameter vectors as input: a starting $\theta$ and a noised future parameter vector $\theta_j^*$, where $j$ denotes the timestep in the diffusion forward noising process. We minimize the simplified variational lower bound, which reduces to predicting the denoised future parameters:

$$\mathcal{L}(G) = \mathbb{E}\left[||\theta^* - G(\theta_j^*, \theta, \ell^*, \ell, j)||_2^2\right] \tag{1}$$

Algorithm 2 details our full training procedure. Note that we need tuples of data $(\theta^*, \theta, \ell^*, \ell)$ to compute $\mathcal{L}$. We sample these tuples from our checkpoint dataset. First, we sample a training run uniformly at random. Then, $\theta$ and $\theta^*$ are sampled uniformly at random from the checkpoints saved within the selected training run. We enforce that $\theta$ is always from an earlier training step than $\theta^*$. Note that $\theta$ and $\theta^*$ can be arbitrarily distant, even the initial and final checkpoints from a run.

**Sampling.** After pre-training, we sample updated parameters $\theta^*$ by querying $G$ with an input parameter vector $\theta$, its loss/error/return $\ell$ and a *prompted* loss/error/return $\ell^*$. Sampling begins by feeding-in Gaussian noise as the $\theta^*$ input and gradually denoising it. We use DDPM sampling.

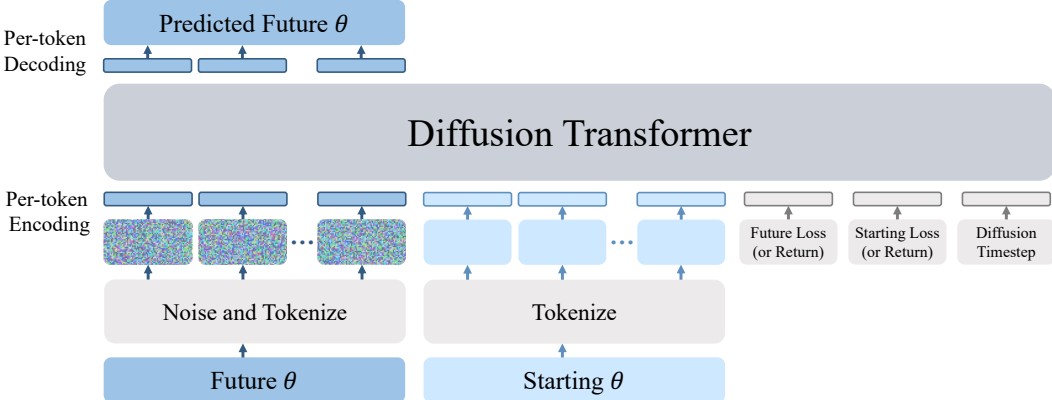

Figure 2: **The G.pt architecture.** During training, we sample two checkpoints from the same run—a "starting" network's parameters and a "future" network's parameters from later in the run—as well as their losses/errors/returns. Each layer's parameters are flattened and linearly encoded. The future network's parameters are noised via a diffusion forward process prior to encoding.

### 2.2.2 ARCHITECTURE

Our generative model is a transformer (Vaswani et al., 2017) that operates over parameter tokens from both $\theta$ and $\theta^*$ (Figure 2). It uses few domain-specific inductive biases beyond tokenization.

**Parameter tokenizers.** Before being processed by G.pt, the two input parameter vectors $\theta$ and $\theta_j^*$ each need to be decomposed into several tokens. In general, a task-level network $f_\theta$ will contain many unique layers, each with a potentially different number of parameters. We define the $i$-th token as the flattened parameter vector of the $i$-th layer. Layers with multiple parameter groups (e.g., layers with both a weight and a bias) are decomposed into separate tokens. Note that these tokens will usually be of different dimensionality. We call this *layer-by-layer* tokenization.

**Parameter tokenizers for big neural networks.** For larger networks, we find that it is beneficial to decompose a single layer's parameters into multiple tokens. We do this with *layer chunking*. We define a hyperparameter $M$, the maximum number of parameters a single token can have. Layers containing more than $M$ total parameters are flattened and chunked into multiple tokens, each with at-most $M$ parameters. For example, if $M = 1000$, a weight matrix with $10 \times 768$ parameters will be decomposed into eight tokens, seven containing 1000 parameters and one containing 680 parameters. We set $M$ to be smaller than the hidden size of the Transformer to avoid lossy compression.

**Metric tokenizers.** We also feed the scalar input metrics $\ell$ and $\ell^*$ (loss, error, return, etc.) and diffusion timestep $j$ as individual tokens to the transformer. We project each scalar to a vector representation using a frequency-based encoding scheme (Mildenhall et al., 2020).

**Per-token encoders.** After tokenizing $f_\theta$'s layers and the input scalars, we project each token to the hidden size of the transformer. We explored more complicated encoders, but find that a simple linear layer works well. Each token's encoder has a unique set of weights.

**Transformer.** The core of the G.pt architecture is a transformer which operates on the set of input parameters and metrics, linearly-encoded into tokens. Our transformer is a version of GPT-2 (Radford et al., 2019). We omit causal masking as our model is not autoregressive across tokens.

**Per-token decoders.** The final layer of G.pt is a decoder from the transformer's output to the future parameter vector. The $i$-th token is linearly decoded from the transformer's hidden size back to the original size of the $i$-th layer's flattened parameter vector. Note that only the output tokens for the noised future parameter vector $\theta_j^*$ are decoded to predictions. Our decoders do not share weights.

**Global residual connection.** Finally, we find that it is beneficial to add a residual connection (He et al., 2016) to the input $\theta$ at the very end of G.pt. This amounts to predicting the parameter *update* $\theta^* - \theta$ instead of directly predicting $\theta^*$ itself. This residual connection also allows us to initialize $G$ to perform the identity function by initializing the decoder weights to zero. Empirically, the global residual connection in conjunction with the identity initialization significantly accelerates training.

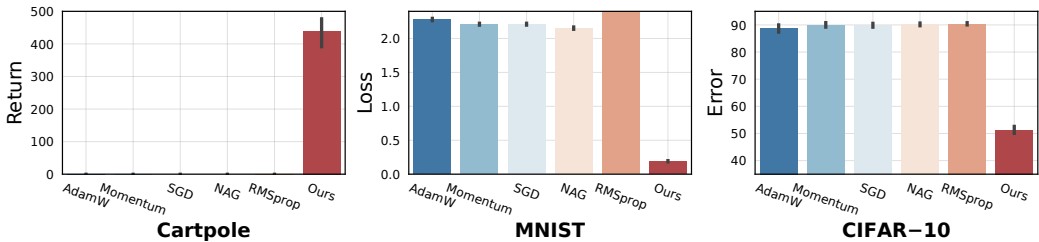

Figure 3: **G.pt optimizes unseen network parameters in one step.** We compare performance after a single update from G.pt versus a single step of gradient-based optimizers. Error bars are computed over five input parameter vectors, all of which are randomly-initialized.

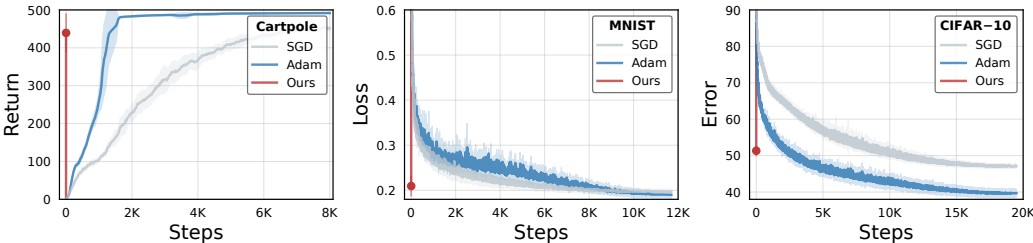

Figure 4: **Optimization curves.** We compare one step of G.pt optimization to training curves produced by SGD and Adam. Error bars are computed over five initializations.

## 3 IMPLEMENTATION DETAILS

We consider both supervised and reinforcement learning tasks. In all cases, we generate a large collection of training runs in order to pre-train a generative model.

**Pre-training data for supervised learning.** We create datasets of MNIST and CIFAR-10 network checkpoints. For MNIST, the task-level model is a two-layer MLP with 10 hidden units; for CIFAR-10, the model has two conv layers followed by global average pooling and a fully-connected layer. Both models use ReLU activations. We train the MNIST models for 25 epochs and CIFAR-10 models for 50 epochs, each with half-period cosine annealing. We use SGD with momentum of 0.9, a learning rate of 0.1 and a weight decay of 5e-4. We train approximately 10K MNIST models and 55K CIFAR-10 models from different random initializations. We select 200 checkpoints to save each run: the initial checkpoint (before training), the final checkpoint and intermediate checkpoints at random iterations. In total, this results in 2M trained MNIST MLPs and 11M trained CIFAR-10 CNNs.

**Pre-training data for reinforcement learning.** For our reinforcement learning (RL) experiments, we train policies for the Cartpole task using the IsaacGym simulator (Makoviychuk et al., 2021). Our policy is a three-layer MLP with 32 hidden units and SeLU activations. We also train a separate critic network with the same architecture as the policy; we only model the policy's parameters in our G.pt experiments. We train for 500 iterations using PPO (Schulman et al., 2017) and Adam (Kingma & Ba, 2015) with $\beta_1 = 0.9$ and $\beta_2 = 0.999$. We train 50K models and record 200 checkpoints in each. This results in a dataset of 10M trained policies.

**Model pre-training.** We train G.pt with AdamW (Loshchilov & Hutter, 2017). We maintain an exponential moving average (EMA) of G.pt weights over the course of training. Our transformer uses a hidden dimension between 1536 and 2048 depending on dataset and has 12 hidden layers with 12-16 heads for self-attention. We use learned positional embeddings across all tokens, initialized to zero. We show full G.pt hyperparameters in Table 1 and dataset information in Table 2. We train one G.pt model per-metric, dataset and architecture (e.g., an error-conditional MNIST MLP model).

**Parameter normalization.** We follow DALL·E 2's (Ramesh et al., 2022) normalization scheme, where the data is scaled such that the variance of the marginal distribution matches the variance of ImageNet pixels scaled to [-1, 1], for which diffusion hyperparameters have been tuned. We find that this normalization ensures the forward noising process destroys nearly all signal in $\theta_J^*$; the KL divergence against a standard normal is roughly $8 \times 10^{-6}$ bits/dim across our experiments.

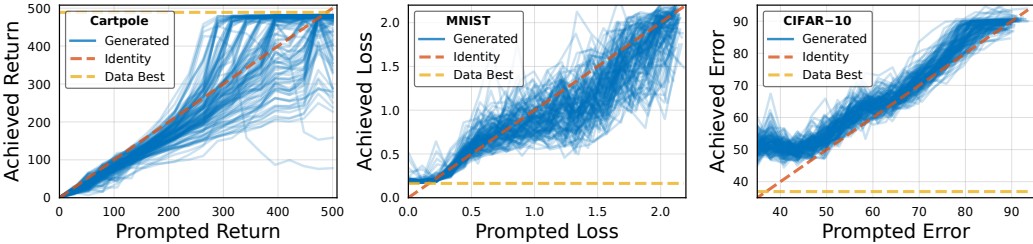

Figure 5: **Achieved returns, losses and errors across a range of input `G.pt` prompts.** G.pt can train unseen neural network parameters to a range of desired values in one update. Each blue curve corresponds to a different randomly-initialized input parameter vector. We also show the best value of each metric present in the training split of the checkpoint dataset.

## 4  EXPERIMENTS

We compare our method to hand-designed optimizers and study the properties of our approach. In all experiments, we report optimization of G.pt on unseen neural network parameters.

### 4.1  COMPARISON TO HAND-DESIGNED OPTIMIZERS

**Training in one step.** Figure 3 demonstrates G.pt's ability to train unseen neural network parameters in one update. This property is unique compared to gradient-based optimizers like SGD and Adam which usually require thousands, if not millions, of updates to achieve good performance. We compare against several of these traditional optimizers with tuned learning rates and weight decays[2]. Note that we did not systematically tune training hyperparameters for checkpoints in our dataset. For each method, we measure performance after applying one update to randomly-initialized network parameters and average results over five seeds. We prompt G.pt by setting $\ell^*$ near the best return/loss/error in our dataset (for some tasks, asking for a value slightly above or below the best value in the dataset works better). G.pt outperforms gradient-based optimizers in this regime across tasks (control, image classification), datasets (Cartpole, MNIST, CIFAR-10) and conditioning metrics (return, test loss, test error). Additionally, G.pt successfully optimizes non-differentiable metrics (CIFAR-10 test error) whereas baseline optimizers must use smoothed surrogates.

**Training in multiple steps.** We compare one step of G.pt to multiple steps of SGD and Adam in Figure 4. SGD and Adam use tuned learning rates and weight decays. The baseline optimizers require thousands of iterations to match the performance of one step of G.pt. With tuned hyperparameters and a sufficiently large number of updates, gradient-based optimizers supersede one-step G.pt optimization. Our model can also be used as an iterative optimizer with recursive prompting. In this setting, we repeatedly feed G.pt's predicted $\theta^*$ back in as its input $\theta$ and ask for low loss/error or high returns. Interestingly, we find that the best performance is usually realized with one-step prompting (recursive prompting usually brings only minor improvements). However, we find that recursive prompting leads to considerably better results when the input neural network comes from an out-of-distribution initialization algorithm not present in our checkpoint dataset (see Figure 6 below).

### 4.2  PROMPTING FOR LOSSES, ERRORS AND RETURNS

By prompting for various desired losses, errors, or returns, G.pt can sample different parameter updates that achieve a range of performance levels. In Figure 5, we show that G.pt successfully learns to generate parameters corresponding to a large range of prompted values. We pass G.pt randomly-initialized neural network parameters and ask it to optimize them in one step to a range of losses/errors/returns. We show results for several different starting parameters. Across different tasks and metrics, G.pt generates parameter updates that are well-correlated with the prompted value. While our model is able to achieve a range of prompted values, we note that it currently shows limited ability to extrapolate to values beyond the limits of the pre-training dataset.

---

[2]We perform a grid search over three learning rates (the PyTorch default and $10\times$ above/below) and three weight decay values ($0, 5 \times 10^{-5}, 5 \times 10^{-4}$) for each baseline optimizer.

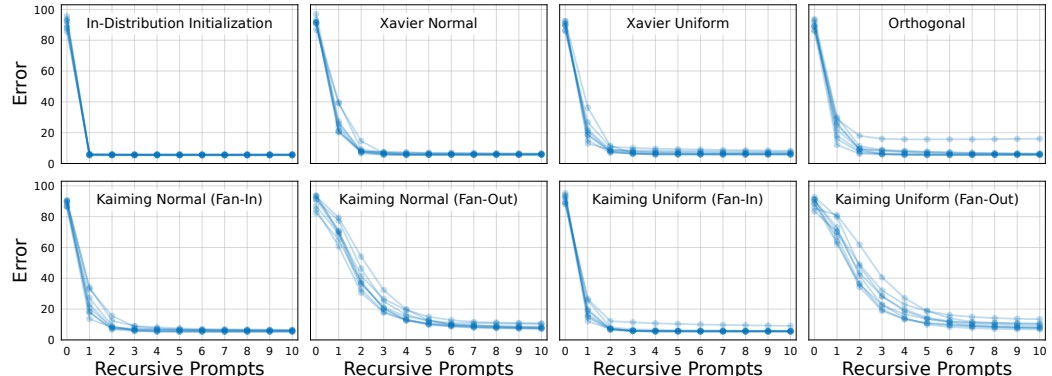

Figure 6: **G.pt generalizes to out-of-distribution parameter initializations.** We query G.pt with randomly-initialized weights sampled from a different distribution than those in our MNIST checkpoint dataset. By recursively applying G.pt to its own output and prompting for low test error, we rapidly optimize out-of-distribution random initializations.

### 4.3 GENERALIZATION TO OUT-OF-DISTRIBUTION INITIALIZATIONS

The networks in our checkpoint dataset are initialized with a single weight initialization scheme. For MNIST, they are sampled $\theta \sim U[-\frac{1}{\sqrt{n}}, \frac{1}{\sqrt{n}}]$, where $n$ is the fan-in of a layer. In Figure 6, we evaluate G.pt's ability to generalize to randomly-initialized input parameter vectors $\theta$, where the weights are sampled from different distributions (Glorot & Bengio, 2010; Saxe et al., 2013; He et al., 2015) not present in our dataset. While one step prompting performance is degraded, recursive prompting significantly improves results. G.pt is able to rapidly optimize out-of-distribution weights in ten or fewer parameter updates.

### 4.4 SCALING MODEL AND DATA SIZE

**Performance metric.** We use prompt alignment to measure scaling performance. We define it as the $R^2$ coefficient of determination between a set of input loss/error/return prompts and the actual loss/error/return achieved by the parameters sampled from G.pt. We compute $R^2$ values over 20 regularly-sampled prompts and average results over 128 neural networks. The optimal score is $+1$, which indicates that G.pt perfectly listens to loss prompts. Randomly-initialized G.pt score around $-2.7$. We use unseen, randomly-initialized input networks in order to gauge generalization capabilities. Empirically, we find that prompt alignment is a more reliable quality metric than diffusion mean-squared error on unseen parameter vectors (see Appendix C for additional details).

**Model scale.** We analyze the impact of increasing the number of G.pt parameters in Figure 7 (top). We train six models with transformer hidden sizes in [64, 128, 256, 512, 1024, 2048]; the smallest model is approximately 2M parameters while the largest is 858M parameters. We evaluate the G.pt checkpoint that attains the highest prompt alignment score over training. We find that larger models generalize much more effectively than smaller models. Small models (<60M parameters) largely fail to generalize to unseen parameter vectors. Even at roughly $10^9$ parameters, we find that G.pt has not saturated its model scaling curve.

**Data scale.** Next, we analyze the impact of increasing the number of training checkpoints in Figure 7 (bottom). We train our largest 858M parameter model on [500, 5K, 10K, 25K, 55K] runs, with each run containing 200 checkpoints. Performance improves substantially as the number of training checkpoints is scaled from 100K to 5M. We do not observe significant improvement when further increasing from 5M to 10M checkpoints. This may be a result of G.pt requiring additional model scale to benefit from a larger pre-training dataset.

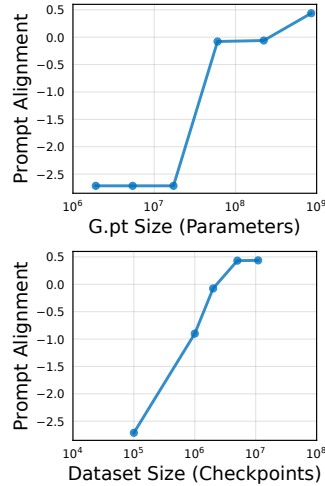

Figure 7: **Scaling studies.**

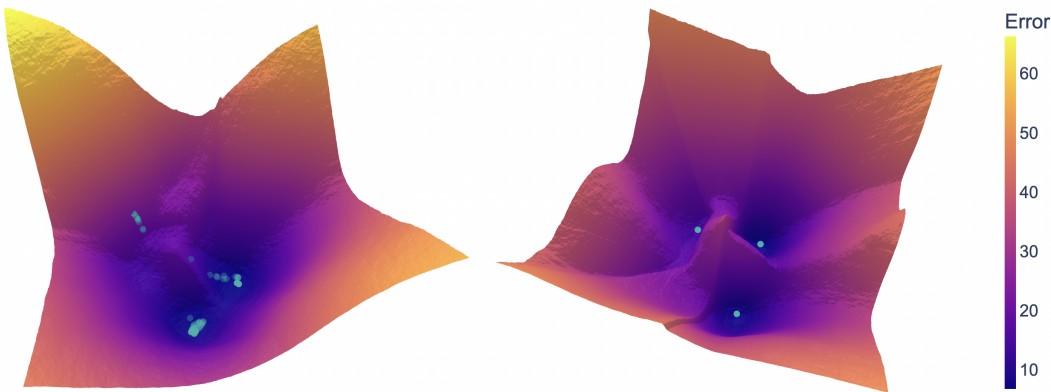

Figure 8: **`G.pt` learns a multimodal distribution over local error minima.** We visualize the test error landscape for an MNIST MLP via parameter space PCA directions (Li et al., 2018). The dots are samples from `G.pt` when prompted for low test error; the two plots use different MLP initializations. With fixed inputs, `G.pt` samples diverse solutions that cover distinct positive-curvature regions of the error landscape. We show `G.pt` samples that reconstruct accurately from PCA encoding.

### 4.5 DIVERSITY OF GENERATED PARAMETERS

The mapping of loss values to neural network parameters is one-to-many. As a generative model, `G.pt` is able to sample diverse parameter solutions given a loss prompt. By fixing all `G.pt` inputs (including $\theta$) and varying sampling noise, we can sample multimodal solutions that cover distinct error minima (Figure 8). Visual inspection of generated first-layer weights suggests that sampling noise controls subtle variations in individual filters as well as the specific ordering of filters.

Intuitively, conditioning on a starting $\theta$ should narrow the space of possible parameter solutions (in other words, initialization should have some bearing on where optimization converge). Indeed, we find that the most significant variation is obtained by re-sampling the starting $\theta$. See Appendix A for a more thorough discussion and additional visualizations.

### 4.6 DATASET DESIGN DECISIONS

**Parameter augmentation aids generalization.** In the absence of permutation augmentation, we observe that `G.pt` can aggressively overfit the training set and fail to generalize to new networks (i.e., it exhibits poor prompt alignment when taking unseen networks as input). Figure 11 (Appendix C) shows that training with parameter augmentation alleviates overfitting in our Cartpole `G.pt` model.

**Training on intermediate checkpoints improves one step training.** Given the redundancy in neural network parameters over a single training run, it is worth asking if there is value in training `G.pt` on 200 intermediate checkpoints per-run. Instead, we could train `G.pt` exclusively on the initial and final checkpoint from each run. We find that this setup degrades one step training capabilities by over 50%: average test loss when prompting with $\ell^* = 0$ worsens from 0.2 to 0.32. `G.pt` significantly benefits from training on a large number of checkpoints, even those from the same run.

## 5 RELATED WORK

### 5.1 PRE-TRAINING FROM LARGE-SCALE DATA

**Transformers for X.** Transformers (Vaswani et al., 2017) were initially developed for language but have been shown to be well-suited for a wide range of domains. They have achieved strong results in vision (Dosovitskiy et al., 2020), language modeling (Radford et al., 2018; 2019; Brown et al., 2020), coding (Chen et al., 2021b; Li et al., 2022), , reinforcement learning (Chen et al., 2021a; Janner et al., 2021), image synthesis (Esser et al., 2020; Ramesh et al., 2021; Yu et al., 2022) and protein folding (Jumper et al., 2021). Likewise, we show that transformers can be used for learning to learn by generative pre-training from neural network parameters.

**Diffusion.** Diffusion models (Sohl-Dickstein et al., 2015) have recently been shown to be highly effective for images (Ho et al., 2020; Nichol & Dhariwal, 2021; Dhariwal & Nichol, 2021; Ho et al., 2021; Ramesh et al., 2022; Saharia et al., 2022; Song & Ermon, 2019). In this paper, we show that diffusion models can be used for meta-learning by generating neural network parameters.

**Pre-training.** Large scale pre-training has led to significant advances in vision (Krizhevsky et al., 2012; Girshick et al., 2014), natural language processing (Devlin et al., 2019; Radford et al., 2018; 2019; Brown et al., 2020) and audio understanding (Van Den Oord et al., 2016; Dhariwal et al., 2020). We explore pre-training from datasets of neural networks instead of datasets of images and text.

**Datasets of neural networks.** Past works have constructed datasets of neural networks and used them in various settings: analyzing population-level trends (Radosavovic et al., 2019; 2020), benchmarking neural architecture search (Ying et al., 2019), training hypernetworks (Knyazev et al., 2021), predicting model properties (Schürholt et al., 2021) and dataset distillation (Wang et al., 2018; Cazenavette et al., 2022). We share the goal of using datasets of neural networks, but for the novel meta-learning approach of pre-training a generative model from trained neural network checkpoints.

## 5.2 LEARNING TO LEARN

**Learning optimizers.** Past works have explored parameterizing optimization update rules with neural networks in place of hand-designed rules like Adam. These rules can be parameterized implicitly as neural networks that take gradients as input and output an improved parameter update. They are typically trained with unrolled optimization (Hochreiter et al., 2001; Younger et al., 2001; Gregor & LeCun, 2010; Andrychowicz et al., 2016; Ravi & Larochelle, 2017; Wichrowska et al., 2017; Lv et al., 2017; Metz et al., 2019; 2022) or reinforcement learning (Li & Malik, 2016; 2017).

**Hypernetworks.** Rather than parameterizing update rules, neural networks can be used to directly output or modify other neural networks' parameters (Schmidhuber, 1992; 1993; Hinton & Plaut, 1987). For example, hypernetworks (Ha et al., 2017) train parameter regression networks end-to-end with the task objective. Hypernetworks have subsequently been extended to support sampling different parameter solutions (Krueger et al., 2017; Deutsch et al., 2019; Ratzlaff & Fuxin, 2019).

**Model-agnostic meta-learning.** MAML learns a parameter initialization that is rapidly adaptable to new tasks (Finn et al., 2017). Subsequent work has built simple probabilistic models over learned MAML initializations (Finn et al., 2018). These methods possess similar characteristics as learned optimizers—they rely on unrolled optimization and require differentiable task-level objectives.

**Learning hyperparameters.** A large body of prior work has explored learning hyperparameters of standard optimizers (Bergstra et al., 2011; Feurer & Hutter, 2019). For example, learning rates, weight decays and weight initializations can all be learned via hypergradient descent (Maclaurin et al., 2015; Baydin et al., 2017; Drucker & Le Cun, 1992), Bayesian optimization (Snoek et al., 2012) and reinforcement learning (Daniel et al., 2016; Xu et al., 2017; 2019; Almeida et al., 2021).

**Learning to learn as pre-training.** In contrast to learned optimizers, hypernetworks and MAML, `G.pt` pre-trains from vast amounts of trained neural network checkpoints. Our method does not backpropagate through task-level losses and, as a result, does not require the task metric being optimized for to be differentiable. This allow us to train with standard generative modeling techniques instead of reinforcement learning or unrolled optimization which can be unstable (Metz et al., 2021).

## 6 DISCUSSION

**Limitations.** The current instantiation of our method has several limitations. First, the model sometimes exhibits signs of underfitting the full loss/error landscape, such as with CIFAR-10. Second, our current `G.pt` models struggle to extrapolate to losses and errors not present in the pre-training data. Third, this paper only pre-trains from single-architecture and single-task data. Finally, this paper considers relatively simple datasets of neural networks with static optimizer hyperparameters.

**Conclusion.** We propose generative pre-training from neural network checkpoints. We show that our approach enables rapid optimization of neural networks across tasks (supervised and reinforcement learning) and metrics (losses, errors, returns). Learning algorithms designed by humans have led to large advancements across different areas of artificial intelligence. We hope that our work serves as a step towards learning learning algorithms from data using modern deep learning techniques.

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

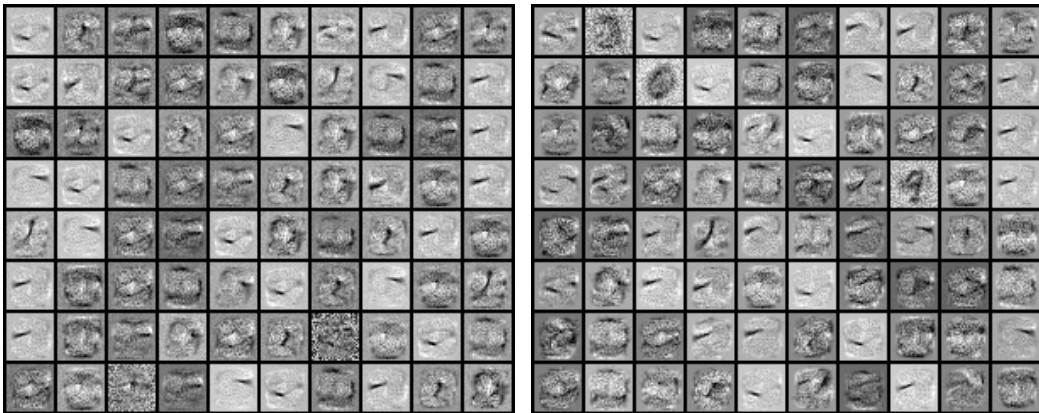

| Generated Parameters | Real Parameters |

Figure 9: **Visualizing synthetic parameters generated by `G.pt` versus real parameters.** Each row visualizes the first layer weights of a single MNIST MLP with 10 hidden neurons. The $j$-th column shows the $28 \times 28$ weights incoming to the $j$-th hidden neuron. *Left:* We show `G.pt` samples when querying for low error. In each row, we feed a different unseen, randomly-initialized input network to `G.pt`. *Right:* The "ground truth" parameters obtained by training the same randomly-initialized nets with gradient-based optimization instead.

## APPENDIX A: ANALYZING MULTIMODALITY

Random noise in the reverse diffusion process and random initialization of the initial network parameters provide two different sources of multimodality in the output parameters for a given target loss/error/return. In our main paper, and also in Supplement Figure 12, we see that `G.pt` is capable of producing multimodal predictions given *fixed* inputs (the input parameters $\theta$, its loss/error/return $\ell$ and the *desired* loss/error/return $\ell^*$). In this section, we show that more variation in `G.pt` is captured by re-sampling input $\theta$. This result suggest that `G.pt` is doing more than memorizing a single network for every input loss/error/return.

**Motivation.** Let's begin by considering the joint distribution over all inputs to `G.pt`:

$$p(\theta^*, \theta, \ell^*, \ell) = p_G(\theta^*|\theta, \ell^*, \ell) \cdot p(\theta|\ell^*, \ell) \cdot p(\ell^*|\ell) \cdot p(\ell) \qquad (2)$$
$$= p_G(\theta^*|\theta, \ell^*, \ell) \cdot p(\theta|\ell) \cdot p(\ell^*|\ell) \cdot p(\ell). \qquad (3)$$

The above follows directly from the probability chain rule and the assumption that $\theta$ is conditionally-independent of $\ell^*$ given $\ell$. Writing the joint distribution in this way makes it clear that there are at least *two* ways we may obtain variation in our generative procedure with fixed inputs $\ell^*$ and $\ell$: we can re-sample $\theta^* \sim p_G$ or re-sample $\theta \sim p(\theta|\ell)$. Sampling from the latter distribution is somewhat complicated by the fact that, for general $\ell$, we do not have a model of $p(\theta|\ell)$[3]. However, for the common case where the input $\ell = \ell_{\text{random}}$ corresponds to the performance of a random neural network prior to training, we can draw reasonable samples from $p(\theta|\ell_{\text{random}})$ by instantiating new networks with whatever initialization algorithm we are using—e.g., Kaiming (He et al., 2015) or Xavier (Glorot & Bengio, 2010) initialization.

In short, given inputs $\ell^*$ and $\ell$, we can produce diverse `G.pt` samples in two ways. First, by re-sampling the noise as part of the reverse diffusion process; second—if $\ell$ is the performance of a random net—by re-sampling random initializations of the input $\theta$.

**Assessing variation.** To disentangle the two sources of variation, we use DDIM sampling (Song et al., 2020) in the reverse process with $\eta = 0$. This sampling scheme forces `G.pt` to produce a deterministic mapping of noise to parameters. Hence, by varying input $\theta$ but holding reverse process noise $z$ constant, we can now assess how much variation in `G.pt`'s distribution over $\theta^*$ is controlled by $\theta$. We evaluate the degree of variability induced by the two sources in our MNIST error model:

---

[3]In principle, we could train a second diffusion model to learn this distribution. We leave this to future work.

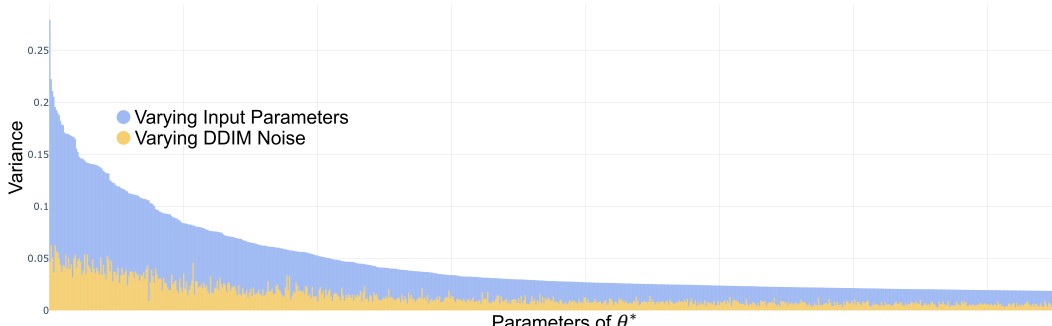

Figure 10: **Varying input $\theta$ yields significant variation in synthesized $\theta^*$.** We compute the per-parameter variance of $\theta^*$ as we re-sample the input $\theta$ vector and hold diffusion reverse process noise fixed (blue). And, we compute the per-parameter variance as we re-sample reverse process noise and hold the input $\theta$ fixed (yellow). The 750 parameters with highest $\theta$-variance are visualized. Input $\theta$ appear to affect G.pt's predictions more strongly than reverse process noise.

1. Diffusion noise re-sampling: $\mathbb{E}_{\theta\sim p(\theta|\ell)}\left[\text{Var}_{z\sim\mathcal{N}(0,I)}(\theta^*)\right]$

2. Input $\theta$ re-sampling: $\mathbb{E}_{z\sim\mathcal{N}(0,I)}\left[\text{Var}_{\theta\sim p(\theta|\ell)}(\theta^*)\right]$

Where $\theta^* \sim p_G(\theta^*|\theta,\ell^*,\ell)$, and both the expectation and variance are taken elementwise per-parameter. We fix the inputs $\ell^*$ and $\ell$. Both the expectations and variances are evaluated with 64 samples, requiring a total of $64 \cdot 64 = 4096$ samples. The results are in Figure 10—indeed, $\theta$ appears to have a stronger effect on G.pt's output than reverse process noise. This result suggests that G.pt does not merely memorize a single solution for every corresponding target loss/error/return $\ell^*$; its predictions appear to strongly depend on the input $\theta$.

## APPENDIX B: ADDITIONAL RESULTS

**Parameter visualizations.** In Figure 9, we visualize G.pt's predicted weights for MNIST MLPs and compare them to real weights from our checkpoint dataset. Varying both input $\theta$ and DDPM sampling noise yields visually noticeable variation in predicted $\theta^*$.

**Error surface visualizations.** In Figure 12, we show nine additional visualizations of the MNIST test error surface and where G.pt samples lie on it. Note that the results are uncurated. All of the plots suggest that G.pt can sample multimodal solutions given fixed inputs $\theta$, $\ell^*$ and $\ell$.

## APPENDIX C: PRE-TRAINING CURVES AND HYPERPARAMETERS

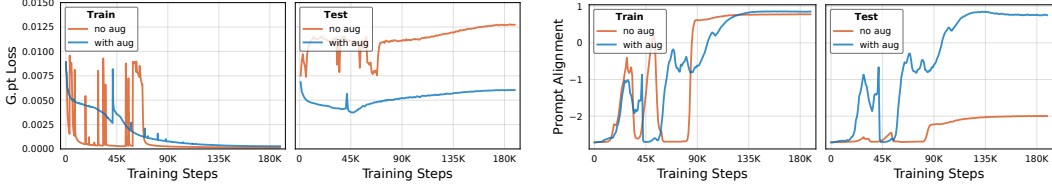

Figure 11: **G.pt training curves with and without parameter augmentation.** *Left:* We show G.pt's loss over the course of training on different splits from our Cartpole checkpoint dataset. *Right:* Prompt alignment scores over the course of training (+1 indicates perfect alignment).

**Training curves.** In Figure 11, we plot various G.pt evaluation metrics over the course of training. Parameter augmentation increases training stability and yields significantly improved generalization as measured by prompt alignment on unseen neural networks in the test set. Interestingly, we sometimes observe that G.pt's prompt alignment on the test set improves while its test loss worsens. Similar trends have been found in autoregressive generative models; e.g., AlphaCode's (Li et al., 2022) solve rate for unseen coding problems improves even as its validation loss increases.

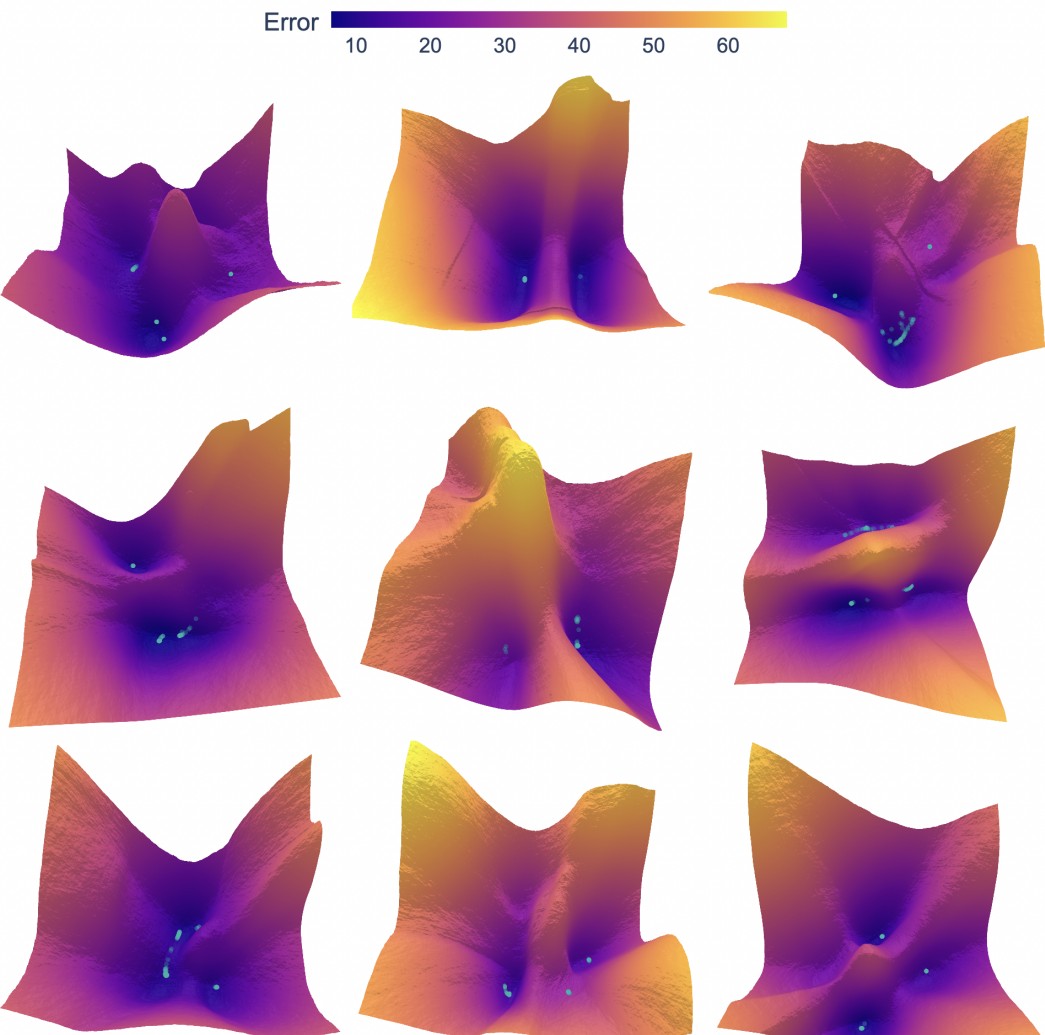

Figure 12: **Uncurated error surface plots.** All nine surface plots visualize the test error landscape of an MNIST MLP. Different plots correspond to different random initializations of the MLP that are input to G.pt. The dots are samples produced by G.pt when prompted for low error. We visualize the surface by moving in the top-two PCA directions computed over 256 G.pt samples. We show G.pt samples that reconstruct accurately from two-dimensional PCA encoding (less than $10^{-3}$ reconstruction distance in parameter space). The plots suggest that G.pt has learned a multimodal distribution over parameters, even when given fixed inputs.

| G.pt Hyperparameters | Cartpole | MNIST (Loss) | MNIST (Error) | CIFAR-10 (Loss) | CIFAR-10 (Error) |
|---|---|---|---|---|---|
| Diffusion steps ($J$) | 1000 | 1000 | 1000 | 1000 | 1000 |
| Noise schedule | linear | linear | linear | linear | linear |
| Transformer layers | 12 | 12 | 12 | 12 | 12 |
| Transformer dim | 1536 | 1536 | 1536 | 2048 | 2048 |
| Self-attention heads | 12 | 16 | 16 | 16 | 16 |
| Model parameters | 347M | 378M | 378M | 639M | 639M |
| Layer tokenizer | layer-by-layer | layer chunk | layer chunk | layer chunk | layer chunk |
| Max parameters per token ($M$) | - | 1000 | 1000 | 576 | 576 |
| Scalar encoder: number of frequencies | 128 | 128 | 128 | 128 | 128 |
| Scalar encoder: max frequency ($\log_2$) | 14 | 14 | 14 | 14 | 14 |
| Data scale factor | 2.06 | 4.185 | 4.185 | 1.646 | 1.646 |
| AdamW $\beta_2$ | 0.999 | 0.999 | 0.95 | 0.999 | 0.999 |
| Base learning rate | $2 \times 10^{-4}$ | $4 \times 10^{-4}$ | $4 \times 10^{-4}$ | $4 \times 10^{-4}$ | $4 \times 10^{-4}$ |
| Learning rate warmup | linear | linear | linear | linear | linear |
| Learning rate decay | cosine | cosine | cosine | cosine | cosine |
| Weight decay | 0.1 | 0.1 | 0.1 | 0.1 | 0.1 |
| Batch size | 8192 | 1024 | 512 | 550 | 550 |
| Gradient clip | - | 0.75 | 0.75 | 0.1 | 0.1 |
| EMA decay | 0.9999 | 0.9999 | 0.9999 | 0.9999 | 0.9999 |
| Training iterations to best performance | 128K | 110K | 288K | 425K | 792K |
| Prompted $\ell^*$ for one-step optimization | 500.0 | 0 | 5.0 | 1.2 | 35.0 |

Table 1: **G.pt Hyperparameters**. When prompting G.pt with $\ell^*$ for one-step optimization, we choose a value close to the smallest loss/error or highest return present in our checkpoint dataset.

| | MNIST | CIFAR-10 | CARTPOLE |
|---|---|---|---|
| #RUNS | 10728 | 56840 | 50026 |
| #TRAIN RUNS | 10228 | 54790 | 47976 |
| #TEST RUNS | 500 | 2050 | 2050 |
| CHECKPOINTS/RUN | 200 | 200 | 200 |
| #CHECKPOINTS | 2.1M | 11.3M | 10M |
| ARCHITECTURE | MLP | CNN | MLP |
| #PARAMETERS | 7960 | 5370 | 1250 |

Table 2: **Checkpoint dataset statistics**.

## APPENDIX D: OVERFITTING TEST SET METRICS

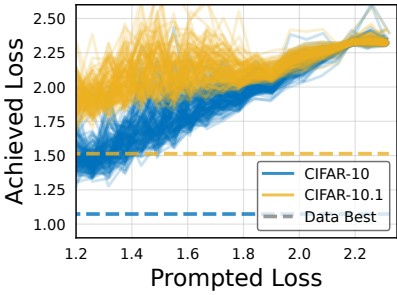

Figure 13: **G.pt pre-trained on CIFAR-10 test losses generalize to CIFAR-10.1.** We examine if G.pt overfits to CIFAR-10 test losses by evaluating its generated networks on CIFAR-10.1. Prompting for smaller CIFAR-10 test losses yields networks that perform better on CIFAR-10.1.

In our supervised learning experiments, we condition G.pt on test set metrics, like losses and errors. A reasonable concern is that conditioning on test losses might cause G.pt to overfit to the task's test set. To investigate this, we evaluate networks generated by our test loss-conditional CIFAR-10 model on the CIFAR-10.1 dataset (Recht et al., 2018). Results are in Figure 13. Asking for smaller CIFAR-10 test loss yields a smaller CIFAR-10.1 loss, indicating G.pt is not overfitting. As with past works that evaluate on CIFAR-10.1, our generated networks perform worse on 10.1 than 10. This is explained by the observation that checkpoints in our dataset also have worse losses on 10.1 (the best checkpoint in our dataset has a CIFAR-10 test loss of 1.07 versus a CIFAR-10.1 loss of 1.51).

APPENDIX E: MEMORIZATION VERSUS GENERALIZATION

Figure 14: **G.pt predictions on held-out (unseen) random initializations tend to lie closer to the ground truth outcome of SGD/Adam than any parameter vector from our checkpoint dataset's training split.** For each test run in our dataset, we feed the initial parameters and a metric prompt to G.pt, and we sample a prediction. We count the percentage of runs for which the prediction is closer to one of the 200 checkpoints in that same test run than all checkpoints in the training split (Cartpole has 10M training split checkpoints, CIFAR-10 has 11.3M and MNIST has 2.1M). Each plot corresponds to a different G.pt model, and we repeat the test for a wide range of prompts.

In this section, we investigate the extent to which G.pt memorizes solutions from the training set. This is a challenging topic to address for any generative model, and there is no universally-accepted methodology to measure it. For generative models of images, one popular methodology is to visualize the nearest neighbors of generated images in the training set. Visualizing parameters is challenging for deep networks beyond the first layer, so we instead provide a basic way to quantify memorization.

**Experimental setup.** Our approach is also based on nearest neighbors. We feed G.pt an unseen, randomly-initialized parameter vector from a test run and sample a corresponding solution $\theta^*$ from our model. If G.pt is memorizing parameters from the training set, then the sampled $\theta^*$ should be closer to one of the millions of parameter vectors across *all* runs in the training split than the 200 "ground truth" parameter vectors in the same test run from which we took the randomly-initialized input parameters. On the other hand, if $\theta^*$ is closer to one of these 200 held-out checkpoints, it suggests that it is accurately predicting the outcome of gradient-based optimization (i.e., there is some level of meaningful generalization). For simplicity, we compute distances in Euclidean space. We count the percentage of test runs for which G.pt generates a solution closer to any of the 200 checkpoints in the test run than all checkpoints in the training split of our dataset (Cartpole has 10M training split checkpoints, CIFAR-10 has 11.3M and MNIST has 2.1M). We call this percentage the *nearest neighbor score*. A score of 100% suggests G.pt is perfectly generalizing. We repeat this test for a range of loss, error and return prompts.

**Results.** Figure 14 shows nearest neighbor scores for all five of our G.pt models. Our models appear to accurately generalize under a large number of loss, error and return prompts. Our Cartpole and CIFAR-10 models exhibit perfect scores (100%) for all input prompts. Interestingly, while our MNIST models also have perfect scores for the majority of loss/error prompts, they have lower scores for smaller prompts (decreasing to about 15-20%). A speculative explanation is that our MNIST models were trained with about a fifth of the number of training runs compared to our Cartpole and CIFAR-10 models; this could possibly degrade generalization capabilities. Overall, this test provides some initial evidence that G.pt is generalizing and not just memorizing training set parameters.

