# OpenReview forum: "Learning to Learn with Generative Models of Neural Network Checkpoints"
_ICLR.cc/2023/Conference — Submitted to ICLR 2023_

### Official Review · Reviewer_rREn · 2022-10-20

**Confidence:** 3
**Correctness:** 3
**Technical Novelty And Significance:** 3
**Empirical Novelty And Significance:** 2
**Recommendation:** 5

**Clarity, Quality, Novelty And Reproducibility:**

I’m satisfied with the overall writing quality and originality of the work. I raised some concerns about the clarity and the significance in the weaknesses above. I think the work provided sufficient details for reproducing the main results.


**Strength And Weaknesses:**

Strengths:
- This work is clearly written, and the presentations of the proposed idea and experiment settings are overall very easy to follow.
- The idea of training a conditional diffusion model in the (parameter, metric)-space to perform the learning to learn tasks looks novel to me.
- Experiments on a fairly reasonable range of settings (MNIST, CIFAR-10, and Cartpole) demonstrated that the proposed method can learn the distribution of network parameters conditioned on the desired metric (note that its value cannot surpass the best from the dataset). Also, experiments showed some nice properties of the proposed method, including scaling the model and data size improves the performance, measured by the prompt alignment score, and sampling diverse outputs for the same metric.

Weaknesses:
- Although I liked the idea of generative pre-training that learns to optimize the parameters, given the claimed properties and current experimental results, my major concern is whether the proposed method is really useful for solving downstream problems. For instance, if the proposed method cannot extrapolate to the metric values beyond the limits of the checkpoint dataset, what extra value does it add on top of the pre-trained checkpoints? Because in practice, especially the classification and RL tasks, we mostly just care about when and how we achieve the better scores (e.g., accuracies and rewards). The claimed favorable properties also seem not add significant value in this regard. Instead of converging to a better local minima, why do we care about how the network ends up with multiple given “mediocre” local minima from unseen initializations? Can we find some use cases to show the practical significance of the proposed method?
- I realized the authors emphasized many times about “from unseen neural network parameters”. I’m a little confused: Does it just mean we initialize the network parameters that are not from the training set?
- Another confusing part is that the proposed method is claimed to optimize the parameters “in just one update”. I’m not sure what the one update exactly means. If we use the DDPM sampling, which is an iterative process with usually hundreds of function evaluations, how can we say “one update”? When comparing with the conventional SGD methods, I think we should consider the DDPM sampling steps.


**Summary Of The Paper:**

This work proposed a generative pre-training approach, termed G.pt, that learns to optimize the parameters of neural networks. Specifically, this approach first constructs a dataset of network checkpoints (along with their corresponding metrics) sampled from a hundred thousand training runs, and then trains a conditional diffusion model on this dataset to learn the distribution of network parameters that fulfill the desired metric. After training, given the target metric and the randomly initialized (or unseen) parameters, we can find the optimized network parameters by directly sampling from the conditional diffusion model. Compared to the conventional SGD type optimization methods that relies on at least thousands of parameter updates, the proposed method is claimed to perform well with “a single parameter update”.


**Summary Of The Review:**

Overall, I liked the idea but I have major concerns about the practical significance of the proposed method.

---

> ### Author Response · Authors · 2022-11-19
> **Authors' Response**
>
> Thank you for your comments. Please find our response below:
>
> > Can we find some use cases to show the practical significance of the proposed method?
>
> We agree that showing extrapolation and generalizations to new architectures and tasks is what we would like to see ideally. However, we think that our results, while not practically useful yet, are nontrivial and provide evidence that this direction may lead to something practically useful eventually. Please see our [meta response](https://openreview.net/forum?id=JXkz3zm8gJ&noteId=lTFmZ7AXFWH) for discussion.
>
> > ​​Does it just mean we initialize the network parameters that are not from the training set?
>
> That is correct, we mean unseen network parameters. Note that we also evaluate our models with parameters from different initialization distributions (e.g., Xavier, Kaiming, etc; please see Figure 6).
>
> > Another confusing part is that the proposed method is claimed to optimize the parameters “in just one update”.
>
> By “one update” we mean one parameter update, which corresponds to “one sample” from the generative model. We will clarify this in the text.

---

### Official Review · Reviewer_iYN2 · 2022-10-23

**Confidence:** 4
**Correctness:** 4
**Technical Novelty And Significance:** 3
**Empirical Novelty And Significance:** 3
**Recommendation:** 8

**Clarity, Quality, Novelty And Reproducibility:**

- Clarity/quality: The paper was clear and easy to follow, and the quality of the experimental results was quite high. I also liked the additional investigation that the authors did in terms of looking at memorization/generalization, as well as exploring the additional source of randomness in the NN weight initializations.
- Novelty: The approach was quite novel. While the idea of learned optimizers is not new, most people have tried to tackle the problem from directly pre-training on a variety of datasets/tasks. I thought the prompting idea to get the model to target a desired test loss was creative and interesting.
- Reproducibility: The paper included sufficient levels of detail in terms of how to preprocess the inputs, how to construct the dataset of (model parameter, loss) pairs, and hyperparameters for each setting.


**Strength And Weaknesses:**

Strengths:
- The idea is really interesting and the paper was fun to read!
- There were a lot of empirical design decisions that had to be made in order to make the approach work (e.g. how to tokenize each layer of the NN, how to modify the conditional Transformer-based diffusion model, permutation augmentation in weight space to facilitate generative pre-training, etc.) and the paper did a good job outlining each step, as well as justifying the reasons behind each choice.
- The results are impressive. Though one could argue that the tasks are relatively simple, I thought the paper did a great proof of concept that G.pt is a promising approach for learning to learn.

Weaknesses:
- The authors did a good job addressing the limitations of the work in Section 6, where they outlined how G.pt does not quite handle extrapolation of loss/error values not present in the training data very well.
- This is not quite a weakness per se, but I thought Appendix A was interesting in terms of constructing an alternate view of the generative process of the joint over (model parameters, losses). It would’ve been helpful for me for the authors to elaborate upon this discussion point a bit more, either in the main text or in the Appendix. Intuitively, it seems as though the dependence between the loss (\ell) and model parameters (\theta) should go in the opposite direction (e.g. p(\ell \vert \theta), with a prior over model parameters p(\theta)) rather than the other way around, since the loss is a function of the model parameters and the input. Things are a bit different here since the diffusion model is being \emph{prompted} with fixed loss values and initial model parameters. Would the authors comment on this?



**Summary Of The Paper:**

This work trains a generative model (conditional diffusion Transformer) called G.pt over NN checkpoints of supervised + reinforcement learning (RL) tasks such that given a prompt of (initial input parameter vector, target loss/error/return), the generative model outputs an updated parameter vector that achieves the desired metric. The paper explores classification on MNIST and CIFAR-10 for MLPs and CNNs, as well as an MLP on a Cartpole task. This approach performs favorably to existing optimizers – G.pt acts as a learned optimizer such that initially, each step of G.pt corresponds to thousands of gradient update steps of traditional optimizers such as Adam and SGD.

**Summary Of The Review:**

The authors introduce G.pt, a generative model of NN checkpoints as a way to efficiently learn how to optimize NNs to target a desired loss/return value. They propose how to make this idea work well in practice with different approaches of tokenizing each NN layer, encoding the loss/model parameters, performing data augmentation, adapting the underlying diffusion model, and outlining the pretraining procedure. They then empirically explore G.pt’s performance on MNIST and CIFAR-10 classification as well as learning a policy to perform Cartpole, and also demonstrate that the model is not straightforwardly memorizing models it has seen before. The submission is high quality, interesting, and novel. I anticipate this being of broad interest to the community, and therefore recommend acceptance.

---

> ### Author Response · Authors · 2022-11-19
> **Authors' Response**
>
> Thank you for your encouraging comments. Please find our response below:
>
> > It would’ve been helpful for me for the authors to elaborate upon this discussion point a bit more
>
> We believe that the factorization suggested by the reviewer is valid; it would be interesting to explore learning it from our dataset of checkpoints. We choose the factorization with $p(\theta^\star | \theta, \ell^*, \ell)$ since it matches the conditional generative model we trained and allows us to reason about possible factors of variation. We will include additional discussion in text.

---

### Official Review · Reviewer_bACh · 2022-10-26

**Confidence:** 3
**Correctness:** 3
**Technical Novelty And Significance:** 4
**Empirical Novelty And Significance:** 4
**Recommendation:** 5

**Clarity, Quality, Novelty And Reproducibility:**

The paper is well-written, easy to understand, and the proposed method is straightforward. I think this paper has novelty and originality in that it presents a new research topic, 'learning to optimize neural networks'.

**Strength And Weaknesses:**

Strength
1. I am not sure whether this research topic is promising, but it is definitely interesting. Also, I believe this paper can serve as the first step towards this research direction.
2. The proposed method is straightforward, and the experimental results are surprising.
3. Unlike the existing gradient-based optimization algorithms, G.pt can optimize non-differentiable objectives, such as classification errors.

Weakness
1. The proposed method can be used only for small models, and such a model has limitations in minimizing generalization errors. Therefore, when deploying a deep learning model, I do not know when it is better to use G.pt instead of an existing optimizer. Could the authors provide an additional discussion about when the proposed optimizer is beneficial compared to existing optimizers?
2. It is interesting that G.pt can easily find multiple parameter sets with the same loss, but there is no discussion of why they are necessary.


**Summary Of The Paper:**

First, the authors built a dataset consisting of neural network checkpoints that perform specific tasks, such as MNIST classification, including its loss and error. The proposed data-driven optimizer, G.pt, is a neural network model trained using this dataset. Specifically, if the parameters of the initial checkpoint, the initial loss, and the target loss are given as inputs, then G.pt outputs a new parameter set that can achieve the target loss. Finally, by using the MNIST, CIFAR-10, and Cartpole datasets with a small architecture (e.g., two-layer MLP with 10 hidden units), the authors validated the effect of G.pt compared to an existing gradient-based optimizer (e.g., Adam optimizer).

**Summary Of The Review:**

The ideas and research topics of the paper are interesting, and the experimental results are also impressive. However, there are obvious limitations in terms of applicability, so I believe that further discussion is necessary in this regard.

---

> ### Author Response · Authors · 2022-11-19
> **Authors' Response**
>
> Thank you for your comments. Please find our response below:
>
> > Could the authors provide an additional discussion about when the proposed optimizer is beneficial compared to existing optimizers?
>
> While we see evidence that our approach outperforms standard optimizers in tested settings, we believe there is still more work to be done in order for our approach to become the “go-to” optimizer. Please see our [meta response](https://openreview.net/forum?id=JXkz3zm8gJ&noteId=lTFmZ7AXFWH) for discussion.
>
> > It is interesting that G.pt can easily find multiple parameter sets with the same loss, but there is no discussion of why they are necessary.
>
> We will include an additional discussion in a revised version of the manuscript. In short, we believe that being able to model multiple distinct solutions may make generalization to new settings easier.

---

### Official Review · Reviewer_Wayk · 2022-10-27

**Confidence:** 3
**Clarity, Quality, Novelty And Reproducibility:** The writing is clear. The proposed me…
**Correctness:** 3
**Technical Novelty And Significance:** 3
**Empirical Novelty And Significance:** 2
**Recommendation:** 5

**Strength And Weaknesses:**

Pros
- The overall idea is interesting: posing performance-conditional parameters as a conditional generative modeling problem lets us leverage advances in diffusion models and is an interesting alternative to e.g. learned optimizers.

Cons
- There doesn’t seem to be any practical benefit for the method, because your pre-training dataset already includes a lot of checkpoints for your specific setting, and your experiment in figure 5 shows that the method does not extrapolate to unseen losses (no runs improve upon “Data Best”). For raw performance, the result of running G.pt has no benefit over simply using the best checkpoint in the dataset.
- I may be misunderstanding the method, but is anything explicitly preventing this model from simply memorizing checkpoints?

This isn’t a request for additional experiments, but I think evidence of generalization across network architecture or dataset would address my concerns about the method.

**Summary Of The Paper:**

This paper uses a diffusion Transformer to map a pair (initial parameter, target loss) to a distribution over parameters that achieve the specified performance. The model is trained with a pre-training dataset of neural network checkpoints.

**Summary Of The Review:**

My main concern with this paper is that I don't see a practical benefit for the method (see "Strength and Weaknesses" above). I do think the idea is creative and interesting.

---

> ### Author Response · Authors · 2022-11-19
> **Authors' Response**
>
> Thank you for your comments. Please find our response below:
>
> > There doesn’t seem to be any practical benefit for the method
>
> We readily agree that extrapolation beyond the training data and generalization across datasets and architectures would be ideal. However, we believe that generalization to unseen network parameters and metric values is nontrivial and suggest that the proposed approach is worth pursuing. Please see our [meta response](https://openreview.net/forum?id=JXkz3zm8gJ&noteId=lTFmZ7AXFWH) for additional discussion.
>
> > is anything explicitly preventing this model from simply memorizing checkpoints?
>
> This was one of our concerns as well. Our experiments suggest that the model is not simply memorizing the training data and is generalizing. Please see Appendix E for a detailed discussion.

---

### Author Response · Authors · 2022-11-19
**Authors' Meta Response**

We thank the reviewers for their time and effort spent on providing careful reviews. We are glad that all of the reviewers recognize the novelty and appreciate our work (Wayk: “the idea is creative and interesting”; bACh: “definitely interesting”, “the first step towards this research direction”, “experimental results are also impressive”; iYN2: “the idea is really interesting and the paper was fun to read”, “The results are impressive”; rREn: “looks novel to me”, “I liked the idea”).

Moreover, we think that all of the reviewer concerns are valid, and we agree with them. Most notably, we agree that we do not have evidence of extrapolation or that our approach generalizes across datasets and architectures. However, we believe that these questions cannot be covered comprehensively in a single paper, or by a single group, and that a larger community effort is needed to answer these questions.

We believe that our work provides valuable empirical evidence that suggests that the proposed approach is worth exploring. We kindly ask the reviewers to consider our work as a step toward building a larger body of work around data-driven approaches for learning to learn.

---

### Decision · Program_Chairs · 2023-01-20

**Decision:**

Reject

**Justification For Why Not Higher Score:**

The work doesn't have enough support (empirical evidence) to support acceptance.

**Justification For Why Not Lower Score:**

N/A

**Metareview: Summary, Strengths And Weaknesses:**

The paper discusses a data-driven approach for learning to optimize neural networks. By using a dataset of parameter checkpoints the authors train a generative model on the parameters. The authors are able to show that the method can generate parameters for latter stages of training for a given model. The reviewers generally found this an intriguing idea. However, as the authors accept, there is currently no evidence that the method could be used to efficiently learn the parameters on a new problem  -- clearly this is the only thing that would ultimately be useful -- and that therefore the current work might be considered as simply an interpolation approach. The consensus amongst the reviewers was that the work is too preliminary for acceptance in its current form.

**Summary Of Ac-Reviewer Meeting:**

We had reasonable online discussions about this and came to a clear reject consensus in the end.